# TGFβ and the Tumor Microenvironment in Colorectal Cancer

**DOI:** 10.3390/cells12081139

**Published:** 2023-04-12

**Authors:** Maximilian J. Waldner, Markus F. Neurath

**Affiliations:** 1Department of Internal Medicine 1, University Hospital Erlangen, Friedrich-Alexander-Universität Erlangen-Nürnberg (FAU), 91054 Erlangen, Germany; 2Deutsches Zentrum Immuntherapie, University Hospital Erlangen, Friedrich-Alexander-Universität Erlangen-Nürnberg (FAU), 91054 Erlangen, Germany

**Keywords:** colorectal cancer, TGFβ, SMAD4, tumor microenvironment, immune response

## Abstract

Growing evidence supports an important role of the tumor microenvironment (TME) in the pathogenesis of colorectal cancer (CRC). Resident cells such as fibroblasts or immune cells infiltrating into the TME maintain continuous crosstalk with cancer cells and thereby regulate CRC progression. One of the most important molecules involved is the immunoregulatory cytokine transforming growth factor-β (TGFβ). TGFβ is released by various cells in the TME, including macrophages and fibroblasts, and it modulates cancer cell growth, differentiation, and cell death. Mutations in components of the TGF pathway, including TGFβ receptor type 2 or SMAD4, are among the most frequently detected mutations in CRC and have been associated with the clinical course of disease. Within this review, we will discuss our current understanding about the role of TGFβ in the pathogenesis of CRC. This includes novel data on the molecular mechanisms of TGFβ signaling in TME, as well as possible strategies for CRC therapy targeting the TGFβ pathway, including potential combinations with immune checkpoint inhibitors.

## 1. Introduction

Transforming growth factor-β (TGFβ) belongs to a group of more than 30 cytokines and growth factors, which regulate the growth, differentiation, apoptosis, and adhesion of various cell types. Besides its role in embryonic development and tissue homeostasis, TGFβ is involved in the regulation of inflammation, wound healing, and cancer development [1]. In cancer development, TGFβ has opposing roles. On the one hand, TGFβ acts as a tumor suppressor in early stage tumors through the induction of apoptosis or cell cycle arrest in pre-malignant cells [2]. On the other hand, TGFβ has tumor promotion effects, including the regulation of epithelial-to-mesenchymal transition (EMT) and supporting an immunosuppressive tumor microenvironment, which enables tumor immune escape at later tumor stages [1,3]. One of the best studied examples on this dual role of TGFβ in cancer development is colorectal cancer (CRC). CRC is still the third most common cancer worldwide and is frequently diagnosed at advanced stages, which require intense treatment strategies [4].

The molecular characterization of CRC, including the seminal work by the Cancer Genome Atlas Network, led to the distinction of hypermutated and non-hypermutated phenotypes [5]. Non-hypermutated CRC accounts for more than 80% of all CRC cases and is defined by fewer than 8.24 mutations per 10^6^ bases. In contrast with hypermutated cancer, which is categorized by more than 12 mutations per 10^6^ bases, it is mostly characterized by chromosomal instability (CIN)/microsatellite stability (MSS) and the accumulation of driver mutations during tumor progression [5,6]. These mutations affect typical oncogenes or tumor suppressor genes, including *APC*, *TP53*, *KRAS*, *PIK3CA*, *FBXW7*, *SMAD4*, *TCF7L2,* and *NRAS*.

In hypermutated CRC, which accounts for 10–20% of all CRC cases, three quarters are linked to microsatellite instability (MSI CRC) with hypermethylation and silencing of mismatch repair (MMR) genes such as *MLH1.* MSI CRC is frequently associated with a distinct epigenetic phenotype, including the CpG island methylator phenotype (CIMP). The remaining quarter of hypermutated CRCs are caused by mutations in MMR genes or polymerase ε (*POLE*) [6]. Similar to non-hypermutated CRC, mutations in certain driver genes occur with an increased frequency in hypermutated CRC. Among these, mutations in *ACVR2A*, *APC*, *TGFBR2*, *MSH3*, *MSH6*, *SLC9A9*, *TCF7L2,* and *BRAF* have been described [5].

Interestingly, an impairment in the TGFβ pathway can be found in both non-hypermutated and hypermutated CRC. As mentioned above, *SMAD4* (mothers against decapentaplegic homologue 4), one of the downstream signaling mediators of TGFβ-receptor activation, is among the eight most frequently mutated genes in non-hypermutated CRC [5]. In MSI CRC, mutations in the TGFβ receptor type 2 (*TGFBR2*) could be found in more than 50% of cases [5].

The relevance of TGFβ for the pathogenesis of CRC is further highlighted by the consensus molecular subtype (CMS) classification of CRC. As a basis for CMS classification, the transcriptome of more than 4000 CRC samples is evaluated to identify clinically relevant subtypes of CRC based on the gene expression. Overall, four CMSs could be identified. CMS1 accounts for 14% of all CRC cases and is associated with worse survival after relapse. CMS1 is characterized by hypermutation, MSI, CIMP, and frequent mutations in *BRAF.* As the transcriptional profile of CMS1 is specific for immune infiltration and activation, it is designated as “MSI immune” phenotype. CMS2, which accounts for 37%, is named the “canonical” subtype due to the high somatic copy number alterations (SCNA) with the activation of the WNT and MYC pathways. CMS3 is named “metabolic” due to the transcriptional dysregulation of the metabolic pathways. It accounts for 13% of CRC cases and can be further characterized by a mixed MSI status, low SCNA and CIMP, and increased frequency of *KRAS* mutations. CMS4 accounts for 23% of CRC cases and has, similar to CMS2, high SCNA. As a result of the high stromal infiltration together with marked activation of TGFβ-signaling, it is named the “mesenchymal” subtype.

In comparison with all of the other CMS types, CMS4 is associated with inferior relapse-free and overall survival [7]. Together, these data clearly indicate a critical role for TGFβ signaling in CRC.

During recent years, the functional role of TGFβ signaling has been intensely studied in various preclinical models. These data provided a glimpse into the complex and ambiguous role of TGFβ within the development of CRC. Within this review, we will give an overview about the regulation and functional role of TGFβ signaling in CRC with a special focus on the tumor microenvironment.

## 2. Molecular Mechanisms of TGFβ Signaling

The TGFβ protein family contains 32 genes distributed among two subgroups, the TGF and the bone morphogenetic protein (BMP) subfamilies in mammalians. While the TGF subfamily includes TGFβ1, TGFβ2, and TGFβ3; activins A and B; nodal; myostation; and GDFs, the BMP subfamily includes BMP2, BMP4, BMP6, BMP7, BMP9, GDF5, GDF9, and anti-Muellerian hormone (AMH), among others [8]. In mammalians, seven type I, five type II, and 2 type III receptors have been described (for review, see [1,8]). Among these, TGFβRI (also known as ALK5) and TGFβRII are the main receptors for TGFβ. These receptor subunits form a heterotetrameric complex consisting of two type I and two type II receptors. The binding of TGFβ to TGFβRII induces the recruitment of TGFβRI to form a complex with TGFβRI, which mediates further downstream signaling through the serine/threonine kinase activity [9]. This signaling is mainly mediated through proteins belonging to the SMAD (mothers against decapentaplegic homologues) family, which act as transcription factors regulating the proliferation, differentiation, chemotaxis, and immune modulation [1]. Depending on the functional role, three SMAD subfamilies have been described. SMAD1, SMAD2, SMAD3, SMAD5, and SMAD9 belong the receptor-activated SMADs (R-SMADs), which mediate effector function. SMAD4 is designated as the co-mediator SMAD, as it forms complexes with receptor-activated SMADs for transducer effector function. Finally, SMAD6 and SMAD7 are inhibitory SMADs, which suppress TGFβ receptor signaling and thereby act as a negative feedback loop. Overall, TGF signaling via SMAD proteins is regarded as canonical signaling. In addition to SMAD-based canonical signaling, TGFβ-receptor activation can also induce non-canonical pathways, including mitogen-activated protein kinases (MAPK), c-Jun N-terminal Kinase (JNK)/p38 MAPK, extracellular signal-regulated kinases (ERKs), phosphatidylinositol-3 kinase (PI3K)/Akt, rhodopsin (Rho) family GTPases, and TGFβ-activated kinase-1 (TAK1) (Figure 1).

For the remainder of this review, we will focus on the TGFβ-SMAD signaling pathway. This pathway is regulated on all levels, including ligands, receptors, and SMAD molecules (for review, see [8]). For instance, TGFβ is produced by various cell types within the TME, including intestinal epithelial cells, macrophages, or T cells. Thereby, TGFβ is synthesized as a dimeric and inactive precursor protein with a C-terminal sequence named latency-associated peptide (LAP), preventing TGFβ activation. LAP is cleaved in the trans-Golgi and remains attached within the small latent complex (SLC). The SLC then binds to the latent TGFβ binding protein (LTBP) to form the large latent complex (LLP), which is released from the cells and is stored within the extracellular matrix. TGFβ is then activated by the proteolytic cleavage of LAP or the structural modification of LLP, which enable the release of the mature and active TGFβ dimer [1]. For instance, integrin αvβ6 has been shown to promote the release of TGFβ by allosteric binding the prodomain of the latent complex [8]. Similarly, Thrombospondin I (TSP1) activates latent TGFβ through allosteric binding, whereas MMP9 enables TGFβ release through proteolytic cleavage [10]. In the absence of receptor binding, active TGFβ is then cleared from the extracellular space within a short time, thereby enabling tight regulation of TGFβ signaling. Overall, these mechanisms are crucial for the versatile role of TGFβ signaling under physiologic and pathological conditions. In the next section, we discuss how a dysregulation of TGFβ signaling contributes to the malignant transformation of intestinal epithelial cells within the pathogenesis of CRC. 

## 3. Mutations Affecting the TGFβ Pathway as Drivers of CRC Progression

The high frequency of mutations affecting members of the TGFβ pathway clearly indicates the tumor-suppressive effects of TGFβ signaling in CRC [1]. This has been attributed to the direct effects of TGFβ on early CRC cells, where TGFβ induces cell-cycle arrest or apoptosis. Functional data supporting these concepts have been derived from preclinical studies with genetically engineered mice evaluating TGFβ pathway members that are mostly mutated in human CRC [11].

### 3.1. Mutations in TGFβRII

As previously discussed, mutations in TGFβRII are the most frequent mutations of the TGFβ signaling pathway in hypermutated CRC [5]. This is especially true in MSI CRCs, which correspond to group 1 in CMS classification. In these cases, mostly frameshift mutations of the TGFβRII gene have been reported [12]. Thereby, TGFβRII mutations seem to occur late during MSI CRC development, as they are associated with the progression of adenomas to MSI CRC [13]. These data match preclinical studies, which show that TGFβRII mutations are not enough to initiate malignant transformation of the intestinal epithelial cells. However, in combination with mutations of other tumor suppressor genes such as PTEN, APC, TP53 and others, TGFβRII mutations result in tumor progression. For instance, Yu et al. evaluated tumor development in mice with a conditional deletion of TGFβRII, phosphatase, and tension homolog deleted on chromosome 10 (PTEN), or a combination of both in the intestinal epithelial cells [14]. The authors chose these genes because they are frequently mutated in MSI-H CRCs. While the mutations of TGFβRII or PTEN alone did not result in CRC development, the combined deletion of both genes resulted in tumor formation in the small intestine and colon in 86% of mice and metastasis in 8% of tumor-bearing mice. Tumor growth was associated with increased cancer cell proliferation, decreased apoptosis, and decreased expression of cyclin-dependent kinase inhibitors. 

Within the CIN pathway, the sequential accumulation of mutations in critical oncogenes has been reported, which include APC, KRAS, PIK3CA, BRAD, and TP53, and the TGF pathway members SMAD4 and TGFβRII [15]. Similar to MSI-H CRC, Sakai et al. could show, by specifically inducing APC, KRAS, and TGFβRII mutations in intestinal epithelial cell organoids, that the additional acquisition of TGFβRII loss in APC and KRAS mutated organoids resulted in increased metastasis [16]. Together, these data clearly show that TGF signaling was protective against cancer progression, as also indicated by data on the role of SMAD4 in CRC.

### 3.2. Mutations in SMAD4 and Other SMAD Genes

Mutations of SMAD4 are among the most frequent mutations detected in non-hypermuated CRC [11]. Besides mutations of the SMAD4 gene, other mechanisms of SMAD4 silencing have also been described, such as ubiquitylation, sumoylation, or microRNA interference [17]. Again, loss of function mutations of SMAD4 have been associated with late stage disease, including metastasis [18] and poor survival [19]. In preclinical studies, the combination of APC and SMAD4 mutations resulted in a more malignant phenotype of intestinal tumors in mice compared with a single mutation of APC [20]. Similarly, SMAD4 deletion and the activation of Wnt in the intestinal epithelial cells resulted in the rapid formation of dedifferentiated adenomas [21]. These observations have been linked to various downstream mechanisms of SMAD4 signaling. For instance, the loss of SMAD4 in intestinal epithelial cells has been shown to result in the upregulation of chemotactic factors, including Ccl9, CCL15, or CXCL1/8, and thereby induces the recruitment of tumor promoting myeloid-derived cells or tumor-associated neutrophils [22]. Furthermore, SMAD4 deletion has been shown to result in the upregulation of vascular endothelial growth factor (VEGF)-A and VEGF-C to mediate angiogenesis and lymphangiogenesis, respectively [23]. 

Besides SMAD4, mutations have also been reported in SMAD2 and SMAD3 in CRC. However, the incidence of these mutations is much lower in comparison with SMAD4,k and functional data are not available. Among the inhibitory SMADs, several data show a relevant role for epithelial SMAD7 in CRC development. For instance, single nucleotide polypmorphisms (SNPs) have been associated with CRC risk in a study with more than 900 CRC cases [24]. An overexpression of SMAD7 in a non-tumorigenic colon cell line resulted in increased tumorigenicity of these cells after transplantation into athymic nude mice [25]. On a molecular level, SMAD7 prevented TGFβ-dependent Akt phosphorylation and G1 cell cycle arrest, and thereby enabled cancer cell growth while inhibiting apoptosis. In a subsequent study by the same group, SMAD7 overexpression in CRC cells was shown to promote liver metastasis in a murine splenic injection model of CRC [26].

Overall, the currently available data clearly show that TGFβ signaling in cancer cells prevents progression to a more malignant phenotype. This is in contrast with the role of TGFβ in the tumor microenvironment, which we will discuss in the next section.

## 4. TGFβ as a Regulator of the Tumor Microenvironment in CRC

Despite its tumor-suppressive role in the early stages of tumor development, increased TGFβ signaling has been linked to tumor progression in many types of cancer. In CRC, upregulation of TGFβ signaling has been associated with advanced disease. For instance, it has been shown that increased levels of TGFβ1 are correlated with metastatic disease and poor prognosis [27,28]. TGFβ1 has been detected in the hepatic metastasis of CRC and has been proposed as a predictor of metastasis following surgery [29,30]. Further evidence was provided by the CMS classification, where CMS4, which is the characterized activation of TGFβ signaling, is associated with an inferior prognosis regarding relapse-free and overall survival in comparison with CMS1-3 [7]. TGFβ upregulation was further associated with genes indicative for an increased epithelial-to-mesenchymal transition (EMT, angiogenesis, matrix remodeling, complement-mediated inflammation, and stromal infiltration). These data support the concept that the upregulation of TGFβ signaling in late-stage CRC induces a tumor-friendly microenvironment by directly regulating innate and adaptive immune cells, as well as tumor resident cells such as cancer-associated fibroblasts (CAFs), as discussed in the following sections.

### 4.1. Tumor-Associated Macrophages

Tumor-associated macrophages (TAMs) are the main type of immune cells infiltrating into the TME. Undifferentiated M0 macrophages can be polarized into M1 macrophages via classical activation or into M2 macrophages by alternative activation. While M1 macrophages have been associated with a protective role in cancer development due to the release of pro-inflammatory cytokines and cytotoxic substance, M2 macrophages have been described to have tumor promoting effects. These pro-tumorigenic effects of M2 macrophages have been linked to the release of various molecules that exert an immunosuppressive function and thereby inhibit the host anti-tumor immune response (for review, see [31]). TGFβ has been attributed a central role in the recruitment of macrophages and polarization into the M2 phenotype in various types of solid cancer. For instance, TGFβ has been shown to induce integrin and type IV collagenase expression to enhance migration to the stride of inflammation [32]. Kim et al. showed that increased migration of macrophages in response to TGFβ is dependent on RhoA signaling [33]. Furthermore, Arwert et al. found an increased expression of CXCR4 on TAMs in response to TGFβ released by cancer cells [34]. Regarding the induction of a M2 phenotype, a SNAIL-dependent inhibition of a pro-inflammatory phenotype in macrophages has been described [35]. In CRC, the deletion of SMAD4 in cancer cells resulted in a decrease in S100A8+ monocytes in the tumor microenvironment [36]. Furthermore, the release of Collagen Triple Helix Repeat Containing 1 (CTHRC1) by CRC cells induced M2 polarization of TAMs through TGFβ signaling and further promoted hepatic metastasis in mouse models of CRC [37].

In addition to the recruitment and polarization of macrophages by TGFβ, TAMs have been also shown to release TGFβ, which then further promotes tumor progression. For instance, Cai et al. found increased EMT of colorectal cancer cells mediated by TAM-derived TGFβ. EMT was induced through the activation of SMAD/SNAIL signaling in CRC cells [38]. In a recent study, an increased number of CD155+ TAMs were found in clinical CRC samples. These CD155+ TAMs were supporting cancer cell migration and invasion through a TGFβ/STAT3-dependent release of matrix metalloproteinases (MMP) 2 and 9 [39].

Besides the regulation of cancer cells, TGFβ released from macrophages has also been shown to inhibit T cell responses in CRC. For instance, a high number of CD163+ macrophages in colorectal polyps from pediatric patients were associated with a high TGFβ expression and reduced T cell infiltration [40]. Further in vitro experiments showed that CD163+ released high levels of TGFβ to suppress T cell proliferation, proposing an immunosuppressive role for TGFβ released from TAMs in CRC.

### 4.2. Tumor-Associated Neutrophils

Similar to macrophages, neutrophil granulocytes have also been implicated in the pathogenesis of CRC. The first evidence came from observations that high numbers of neutrophils in the peripheral blood of CRC patients were associated with an inferior prognosis. For instance, Rashtak et al. evaluated the peripheral neutrophil to lymphocyte ratio (NLR) of 2546 CRC patients and found an inferior outcome regarding disease-free and overall survival in patients with a high NLR [41]. Further studies could show that a high NLR is associated with a poor outcome following hepatic resection for liver metastasis or a response to treatment with bevacizumab and chemotherapy (for review, see [42]). 

In the tumor microenvironment, recent data have shown that tumor-associated neutrophils (TANs) also show relevant plasticity and can be polarized into the anti-tumorigenic N1 phenotype or a tumor-promoting N2 phenotype [42]. N1 neutrophils are characterized by the secretion of TNFα, reactive oxygen species (ROS), Fas, and others, and thereby exhibit a cytotoxic function. In contrast, N2 neutrophils release arginase, MMP9, VEGF, and various chemokines to support tumor progression. Again, TGFβ has been described to regulate the polarization of neutrophils into the tumor-promoting N2 phenotype [43]. Regarding CRC, epithelial notch has been shown to induce a CMS4 phenotype with TGFβ-mediated neutrophil recruitment into the TME to drive tumor metastasis in a mouse model of CRC [44]. Furthermore, TANs have been described to suppress the adaptive anti-tumor immune response against CRC by suppressing T cell function via TGFβ [45]. Interestingly, TANs did not directly release TGFβ, but rather released MMPs, which enabled the release of latent TGFβ from the TME. Therefore, neutrophils might enable immunosuppression in a TGFβ-rich environment.

### 4.3. Myeloid-Derived Suppressor Cells

Another group of innate immune cells that are involved in the regulation of tumor progression are myeloid-derived suppressor cells (MDSCs). Although these cells of the myeloid lineage share several features with TAMs and TANs, they are still considered as distinct cell populations. MDSCs have been divided into a monocyte group (M-MDSCs) defined as CD11b^+^Ly6G-Ly6Chigh and a polymorphonuclear group (PMN-MDSCs) defined as CD11b^+^Ly6G^+^Ly6Clow in mice (for review, see [46]). As the name indicates, MDSCs are defined by their immunosuppressive function, which is dependent on the generation of reactive oxygen and nitrogen species; immunosuppressive cytokines, such as IL-10 and TGFβ; or the degradation of L-arginine [46]. Furthermore, MDSCs promote the degradation of L-arginine, which is required for T cell proliferation and effector function, through the release of NO synthase (iNOS) and arginase-1 (ARG1) [46,47]. In several types of cancer, high numbers of circulating MDSCs have been correlated with a worse prognosis [48]. Regarding CRC, Nair et al. found an upregulation of TGFβ signaling in circulating M-MDSCs from the peripheral blood of CRC patients [49]. Similarly, a recent study by Gneo et al. evaluated MDSCs in tissue and peripheral blood samples of CRC samples [50]. In this study, increased numbers of M-MDSCs in the peripheral blood and tissue samples were induced by TGFβ and inhibited T cell proliferation via IL-10 signaling. Thus, MDSCs, especially M-MDSCs, seem to be important regulators of an immunosuppressive environment in CRC.

### 4.4. T Cells

Although a high NLR has been associated with a worse prognosis in CRC patients, and thus low numbers of lymphocytes could be indicative of an inferior course of disease; the role of lymphocytes in CRC progression is dependent on the specific lymphocyte subtype. As shown by the group of Jérôme Galon in 2005, the infiltration of early memory and effector memory CD8^+^ T cells into the TME is correlated with a superior prognosis, including disease-free and overall survival in CRC patients [51]. Of note, the characterization of immune infiltration into the TME allows for a more precise assessment of the prognosis of individual patients than standard TNM classification [52]. As a consequence, the consensus Immunoscore was developed for the routine clinical assessment of CRC [53]. In addition to cytotoxic CD8^+^ T cells, Th1 helper T cells and IFNg have also been associated with a superior prognosis in CRC [15,54]. 

In contrast, the presence of Th17 cells, Th22 cells, and regulatory T cells (Tregs) has been associated with a worse prognosis. While Th17 and Th22 cells directly promote tumor progression through the release of tumor-promoting cytokines such as IL-17A or IL-17, Tregs suppress the anti-tumor effector response of Th1 cells and cytotoxic CD8 T cells. However, data shedding light on the role of individual Treg subtypes in CRC revealed that this is not always the case (for review, see [55]). For instance, CD4^+^ CD25^+^ Tregs have been shown to suppress inflammation-associated CRC development through the release of IL-10 [56], and induce CRC regression in the Apcmin/+ mouse model of CRC by regulating the homeostasis of epithelial cells [57]. In a more recent study, *Nrp1*−/− Tregs supported anti-tumor immunity through the release of IFNγ in mouse models of melanoma and head and neck cancer [58]. In contrast, *Nrp+/+* Tregs harbored a critical immunosuppressive function. As Nrp1+ Tregs have been associated with a worse prognosis in CRC, and similar mechanisms might also play a role here [59].

TGFβ is known as an important regulator of T cell survival, activation, and differentiation [10,60]. TGFβ can inhibit the differentiation of both Th1 and Th2 cells. In the case of CD8^+^ T cells, TGFβ has been shown to inhibit cytotoxic function [61]. In contrast with these T cell populations, which are critical for the host anti-tumor immune response, TGFβ has been shown to promote the differentiation of immunosuppressive Tregs and pro-inflammatory Th17 cells [62]. The relevance of TGFβ for T cell homeostasis is further supported by the fact that genetically modified mice with a dominant-negative TGFβBRII and mice with T cell dependent deletion of TGFβBRII develop systemic autoimmunity, including colitis [63].

Evidence for this well-described role of TGFβ for T cell regulation has also been found in CRC. For instance, Tauriello et al. evaluated mice with a conditional deletion of Apc, KRAS, TGFβBRII, and TRP53 in intestinal stem cells [64]. These mice developed metastatic intestinal tumors with high TGFβ activation in TME. Blocking TGF signaling in this model resulted in a potent anti-tumor T cell response with a Th1 phenotype. To directly evaluate TGFβ signaling in T cells of mice, Kim et al. induced a T cell-specific deletion of SMAD4, which resulted in spontaneous tumors in the colon, rectum, duodenum, stomach, and oral cavity [65]. SMAD4-deficient T cells produced high amounts of IL-5, IL-6, and IL-13. Among these, IL-6 in particular is known as an important promoter of CRC progression [66,67]. In contrast, mice with a transgenic overexpression of SMAD7 in T cells are protected against tumor development in the model of azoxymethan (AOM) and dextran sodium sulfate (DSS)-induced colitis-associated cancer (CAC) or following the subcutaneous injection of syngenic MC38 CRC cells [68,69]. However, these animals develop severe colitis characterized by CD8^+^ T cell infiltration and IFNγ production. To further study the role of TGFβ-signaling in Th17 cells, Perez et al. transferred T cells with a conditional deletion of TGFβBRII under the promoter of IL-17A into T- and B-cell deficient Rag1−/− mice and exposed these animals to the AOM + DSS model of CAC [70]. Highlighting the role of TGFβ for Th17 effector function, Th17 with TGFβRII-deletion produced less IL-22 and resulted in less tumor development in murine CAC. Interestingly, TGFβ has not only been described to direct tumor-promoting T cell development in CRC. Wang et al. evaluated tumor-infiltrating lymphocytes (TILs) in the tissue samples of 20 CRC patients [71]. As shown by additional in vitro studies, TGFβ together with IL-4 was required for Th9 enrichment and Th9 cells induced the expansion of cytotoxic CD8^+^ T cells.

Altogether, these data further show the complexity of TGFβ signaling in the TME of CRC and imply a critical role of T cell regulation in the tumor-promoting effects of TGFβ signaling in CRC.

### 4.5. Cancer-Associated Fibroblasts

In addition to tumor-infiltrating immune cells, tissue resident cells such as fibroblasts have also gained increasing attention in recent years. In the pathogenesis of various types of solid cancer, cancer-associated fibroblasts (CAF) have been recognized as important regulators of tumor progression, mediating EMT, angiogenesis, cancer migration and invasion, therapeutic resistance, and others (for review, see [72]). Although most CAFs develop from resident tissue fibroblasts, they can also emerge from fibrocytes, bone-marrow-derived mesenchymal cells, epithelial cells, endothelial cells, stellate cells, and adipocytes [73]. Independent from the type of cancer, TGFβ, which is known to be an important regulator of tissue fibrosis, has been implicated in the differentiation of CAFs. For CRC, this has been shown in a study by Calon et al. [74]. In 345 CRC samples, the authors found an association between a high TGFβ expression with an inferior prognosis. Of note, TGFβ signaling in stromal cells, especially CAFs, was responsible for the TGFβ gene signature that was associated with a poor prognosis. Furthermore, TGFβ induced IL-11 expression and release from stromal CAFs, which supported the metastasis of CRC cells in mice. Similarly, Hawinkels et al. found a TGFβ-dependent feedback loop between CRC cells and fibroblasts. Here, CAFs were activated by TGFβ and subsequently secreted further TGFβ together with proteinases into the TME [75]. Several further publications could show that TGFβ signaling in CAFs supports the invasion and metastasis of CRC [76,77].

Therefore, similar to myeloid cells and T cells, TGFβ signaling is a critical component of the tumor promoting effects of CAFs in cancer and thus could serve as a potential target for therapeutic interventions.

## 5. Therapeutic Modulation of TGFβ Signaling in CRC

Because of the central role of TGFβ signaling in advanced CRC, several strategies have been developed for therapeutic modulation of the TGFβ pathway in TME. These include, but are not limited to, drugs that interfere with TGFβ synthesis, blockers of TGFβ ligand−receptor interactions, or inhibitors of the receptor kinase activity and downstream signaling pathways (for review, see [78]). Several of these drugs targeting TGFβ, TGFβ receptors, and SMADs have been tested in clinical trials on various types of solid cancer, including CRC [78]. For instance, the anti-TGFβ2 antisense oligodeoxynucleotide Trabedersen has been shown to be effective in preclinical studies on pancreatic cancer and was tested in a phase I study (NCT00844064) on various solid cancers, including CRC [79]. The anti-TGFβRII monoclonal antibody Ly3022859 was tested in 14 cases of advanced solid tumors, including 3 cases of CRC (NCT01646203). However, due to cytokine release syndrome, dose escalation was not possible and the maximum tolerated dose could not be determined [80]. Based on these and similar data that showed systemic side effects with anti-TGFβ therapy, a more specific inhibition of TGFβ signaling and a better selection of suitable patient subgroups have been proposed for the treatment of CRC. 

As a consequence of TGFβ-signaling in TME and the suppression of the anti-tumor T cell response, combining TGFβ inhibition with an additional modulation of the immune response seems to be a promising strategy. During recent years, tremendous success has been achieved by enhancing the anti-tumor T cell response through the immune checkpoint blockade in many types of solid cancer. Immune checkpoint inhibition is achieved by blocking inhibitory pathways such as programmed cell death 1 (PD1) or cytotoxic T-lymphocyte-associated protein 4 (CTLA4), e.g., through specific antibodies, in order to enhance cytotoxic T cell function. In CRC, the immune checkpoint blockade has so far only been effective and approved for the therapy of MSI-H CRC cases, in which a strong activation of the adaptive immune system can be observed [81]. Because of the immunosuppressive role of TGFβ in the TME of CRC, additional subgroups of CRC, e.g., CMS4, could also be rendered susceptible for the immune checkpoint blockade. This concept has been supported by the previously described preclinical study of Tauriello et al., where the anti-TGFβRI kinase inhibitor galunisertib induced susceptibility for anti-programmed cell-death protein 1 (PD-1) therapy in a mouse model of metastatic CRC [64]. The first clinical trials are on the way to translate these finding into human disease (NCT03724851) and the results are eagerly anticipated.

## 6. Conclusions

CRC is a typical example of the dual role of TGFβ signaling in cancer. Defective TGFβ signaling in cancer cells, mostly through the inactivation of TGFβRII in MSI CRC or SMAD4 in non-hypermutated CRC, is a frequent event that has been implicated in CRC progression. These mutations suggest an anti-tumor effect of TGFβ signaling and, in fact, TGFβ has been shown to exert an anti-tumor activity through the induction of apoptosis or cell cycle arrest in premalignant cells. On the other hand, TGFβ is critically involved in the regulation of the tumor microenvironment, where it facilitates the polarization of tumor-promoting innate immune cells or cancer-associated fibroblasts, and it inhibits the adaptive anti-tumor immune response (Figure 2).

Overall, the immunosuppressive effect of TGFβ in TME seems to be of critical importance for CRC progression, and thus the therapeutic inhibition of TGFβ has been frequently regarded as a promising strategy for the treatment of CRC. Although previous studies targeting TGFβ in CRC have only provided limited results, especially due to systemic effects, resent data suggest that the inhibition of the TGFβ pathway in selected patients or in combination with the immune checkpoint blockade might be promising strategies. However, more in-depth knowledge about the regulation of different cell populations by the TGFβ signaling is required to identify suitable targets and promising candidates for combination therapies. Furthermore, clinical markers for the identification of the patient subgroups that would benefit from such approaches would be needed. In this regard, additional large-scale multi-omics data, including single cell RNA sequencing or spatial transcriptomics, could pave the way to successfully integrate the modulation of TGFβ in the treatment of CRC.

## Figures and Tables

**Figure 1 cells-12-01139-f001:**
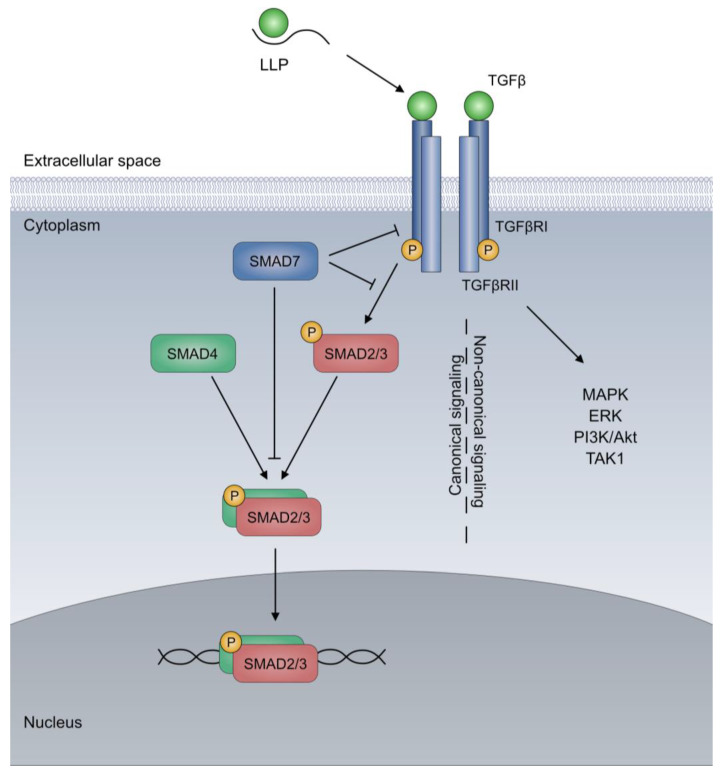
TGFβ signaling pathway. Following the release from the large latent complex (LLP), TGFβ dimers binds to the heteromeric TGFβ receptor, which consists of type I and type II receptors (such as TGFβRI and TGFβII). Within canonical signaling, this results in receptor phosphorylation with the subsequent activation of the receptor activated SMAD molecules (R-SMADs), such as SMAD2 or SMAD3. Phosphorylated R-SMADs form a heteromeric complex with SMAD4, which translocates to the nucleus and induces the transcription of target genes. Inhibitory SMADs (I-SMADs) such as SMAD7 can inhibit TGFβ signaling at several points. In non-canonical TGFβ signaling, TGFβ receptor activation results in the activation of mitogen-activated protein kinase (MAPK), extracellular signal-regulated kinase (ERK), phosphatidylinositol-3 kinase (PI3K)/Akt, TGFβ-activated kinase-1 (TAK1) pathways, and others. The figure has been adapted from [9].

**Figure 2 cells-12-01139-f002:**
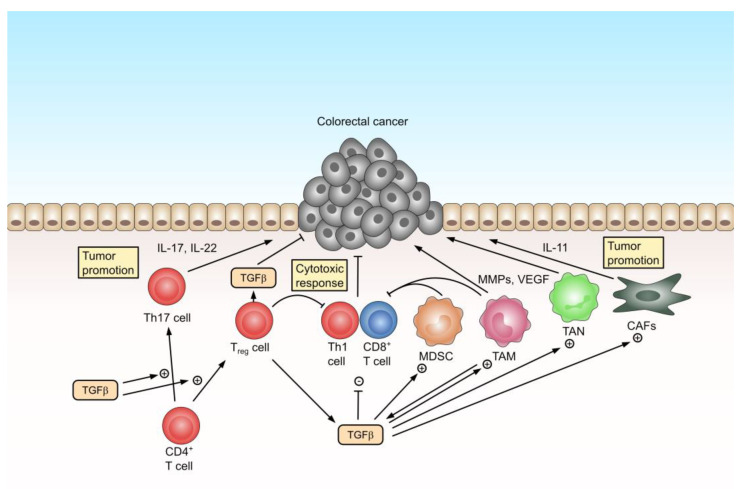
Dual role of TGFβ in the tumor microenvironment of colorectal cancer. TGFβ mediates tumor-promoting effects, e.g., by contributing to a pro-inflammatory environment and suppressing the anti-tumor immune response. Tumor-promoting inflammation is mediated by CAFs or infiltrating TAMs and TANs, which release pro-inflammatory cytokines such as IL-17, IL-22, or IL-11, as well as MMPs and VEGF. TGFβ directly induces Tregs and MDSC to suppress the cytotoxic anti-tumor immune response of Th1 CD4^+^ T cells and CD8^+^ T cells. At the same time, TGFβ inhibits ll proliferation and induces the apoptosis of cancer cells. TGFβ = transforming growth factor β; IL-17 = interleukin-17; IL-22 = interleukin-22; IL-11 = interleukin-11; MDSC = myeloid-derived suppressor cells; TAM = tumor-associated macrophage; TAN = tumor-associated neutrophil; CAF = cancer-associated fibroblast; MMP = matrix metalloproteinase; VEGF = vascular endothelial growth factor.

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
