# Peer review of "TGFβ and the Tumor Microenvironment in Colorectal Cancer"

_cells, 2023, doi:10.3390/cells12081139_

Round 1
Reviewer 1 Report
It is fine to write a review on TGFβ and tumor microenvironment in CRC, but when mentioning the genes mostly affected one should not just state "...including the TGFβ related genes but also the top of the list like TP53, APC and KRAS (which appears at a later position in the review only). Stating that TGFβ related genes are top and mutated in ≈ 30% of cases and then use 2 references more than 20 years old is not appropriate! And in deed newer research shows different numbers and relatively recent publications are available! MSI affects 10-20% of CRC and in this group as stated TGFβRII mutations play a major role.
TGFβRII mutations in CRC are interesting and it is OK to cite some seminal papers like ref. 12 and possibly ref 13 but also add some more recent ones.
There is a lack of consistency in using greek letters throughout the document. Please harmonize!!
Otherwise the review is well written and well structured, thus follows a logical order.
At the end of section 4.5 a sort of concluding remark/statement is missing, which is present in previous sections e.g. in 4.4 "Altogether these data further show the complexity of TGFβ signaling in the TME of CRC and imply a critical role of T cell regulation for the tumor-promoting effects of TGFβ signaling in CRC. " or 4.3 "Thus, MDSCs, especially M-MDSCs, seem to be important regulators of an immunosuppressive environ- ment in CRC. ". So 4.5 seems a bit unfinished in a way.
Author Response
It is fine to write a review on TGFβ and tumor microenvironment in CRC, but when mentioning the genes mostly affected one should not just state "...including the TGFβ related genes but also the top of the list like TP53, APC and KRAS (which appears at a later position in the review only).
We agree with the reviewers comment and have added the most frequently mutated genes in non-hypermutated and hypermutated CRC.
Stating that TGFβ related genes are top and mutated in ≈ 30% of cases and then use 2 references more than 20 years old is not appropriate! And in deed newer research shows different numbers and relatively recent publications are available! MSI affects 10-20% of CRC and in this group as stated TGFβRII mutations play a major role.
We agree with the reviewer that newer data are required. The complete section has been rewritten (see also reply to comments of Reviewer 4).
TGFβRII mutations in CRC are interesting and it is OK to cite some seminal papers like ref. 12 and possibly ref 13 but also add some more recent ones.
Newer literature has been added.
There is a lack of consistency in using greek letters throughout the document. Please harmonize!!
The whole manuscript has been checked to harmonize the use of Greek letters.
Otherwise the review is well written and well structured, thus follows a logical order.
We thank the reviewer for this comment.
At the end of section 4.5 a sort of concluding remark/statement is missing, which is present in previous sections e.g. in 4.4 "Altogether these data further show the complexity of TGFβ signaling in the TME of CRC and imply a critical role of T cell regulation for the tumor-promoting effects of TGFβ signaling in CRC. " or 4.3 "Thus, MDSCs, especially M-MDSCs, seem to be important regulators of an immunosuppressive environ- ment in CRC. ". So 4.5 seems a bit unfinished in a way.
We have added a concluding statement at the end of section 4.5
Reviewer 2 Report
This is a well-written summary of the multiple roles that Tgb signaling can have in tumorigenesis. It is very broad which is both a strength and limitation. The broadness provides selected references for every topic that will permit a reader to gain entry to the much more extensive literature in related to each aspect. This reviewer will likely find it useful for that. However, the broadness precludes in depth discussion or critical evaluation of reports, and especially discrepancies in the literature. Also lacking any sense of what the next critical questions are unanswered. Overall, this is a worthwhile but simplified summary of the broad role that Tgb signaling can play.
Author Response
The reviewer correctly states that our review summarizes the current concepts on the role of TGFβ signaling in the tumor microenvironment of CRC in a rather broad way. We have decided to write our review in this way to give an overview about current concepts and referencing to more specific publications where needed. To improve our manuscript, we have included a paragraph on possible next questions that could enable a successful therapeutic modification of TGFβ signaling in human CRC in the future.
Reviewer 3 Report
Dear authors,
The subject of the review is both important and interesting and is well-written. However, while the abstract briefly touches on the main theme, it lacks sufficient detail to fully convey the major contributions and novelty of the review in comparison to other TME or TGF-beta reviews.
Please update some references to be more current and specific to the discussed sentences. For example, Reference 4 could be updated to Siegel et al. (2023), while Reference 3 could be supplemented with specific papers or reviews that present the duality mentioned.
In addition, the text could be improved by presenting Lines 37-46 as a table with more information about the type of mutation. It would also be interesting to explore whether any evidence exists regarding part of the mutations or epigenetic regulation of TGFbeta signaling being selected in stromal cells, given that the subject is TME.
Furthermore, a brief description of the major differences among subtypes would be informative regarding the information about the TGFbeta being muted more in the CMS4 tumor subtype.
Additionally, a figure could be added to section 2 (Molecular Mechanisms of TGFβ Signaling) to help readers better understand the topic.
Since the theme of the review is TME, it would be valuable to include more information about TGFb activation by proteolytic activity.
Moreover, it is essential to provide references even when previously mentioned in the text. For example, Lines 104-105 specify the mutation as "loss-of-function," but there is no reference to support the statement. Similarly, Lines 107-108 mention functional data supporting the concepts, but again no references are attributed.
Regarding Line 114, there may be new references about the TGF mutation since 1995, which could be included in the text. Additionally, the discussion about ECM and the release of TGFbeta by MMP could be expanded upon, given its relevance to the chosen subject.
Line 118 raises the question of which tumor suppressors are being referred to, as only the ones listed later in the text are mentioned. It would also be beneficial to readers to elaborate on why the degradation of L-arginine is immunosuppressive in Line 250.
Furthermore, Line 226 raises the question of whether "superior predictions" refer to better prognostics. Finally, Line 276 would benefit from further explanations and details regarding effector Tregs (eTreg) and their anti-tumor effects in CRC.
In addition to these points, readers would benefit from a figure summarizing the dual role of TGFbeta, as Figure 1 only depicts its pro-tumoral functions.
The conclusion section could also be expanded to include a brief summary of the major points that support the dual role of TGFbeta and how pharmacological interference could contribute to cancer treatment.
Finally, I note some minor points that could be addressed in the text.
For instance, in the abstract, line 9 "an important role of" or "the central role of" the tumor microenvironment could be used instead of "a central role for " .
Line 14," please add a comma before "etc".
Line 23 should remove the word "are."
Additionally, in Line 117, the parentheses require clarification, and Line 182 requires punctuation at the end of the sentence.
Thank you for considering these recommendations.
Author Response
The subject of the review is both important and interesting and is well-written. However, while the abstract briefly touches on the main theme, it lacks sufficient detail to fully convey the major contributions and novelty of the review in comparison to other TME or TGF-beta reviews.
We thank the reviewer for this comment. More information has been added in the abstract to highlight unique aspects of our review including therapeutic implications.
Please update some references to be more current and specific to the discussed sentences. For example, Reference 4 could be updated to Siegel et al. (2023), while Reference 3 could be supplemented with specific papers or reviews that present the duality mentioned.
Reference 4 has been changed and additional work has been added to reference 3.
In addition, the text could be improved by presenting Lines 37-46 as a table with more information about the type of mutation.
The whole section has been rewritten according to comments of Reviewer 1 and Reviewer 4. As the text now mainly refers to mutations in TGFBRII and SMAD4, a table might not add further information.
It would also be interesting to explore whether any evidence exists regarding part of the mutations or epigenetic regulation of TGFbeta signaling being selected in stromal cells, given that the subject is TME.
Despite intensive literature research, we have not found studies providing data on mutations or epigenetic regulation of TGFβ pathway members in stromal cells. We have added a sentence in the conclusion that more cell specific data on TGFβ signaling in human CRC are required.
Furthermore, a brief description of the major differences among subtypes would be informative regarding the information about the TGFbeta being muted more in the CMS4 tumor subtype.
We have included a description of all CMS subtypes into the introduction.
Additionally, a figure could be added to section 2 (Molecular Mechanisms of TGFβ Signaling) to help readers better understand the topic.
As suggested by the reviewer, a figure has been added to section 2.
Since the theme of the review is TME, it would be valuable to include more information about TGFb activation by proteolytic activity.
We have added a more detailed explanation of TGFβ activation through proteolytic cleavage and allosteric binding of ECM components.
Moreover, it is essential to provide references even when previously mentioned in the text. For example, Lines 104-105 specify the mutation as "loss-of-function," but there is no reference to support the statement. Similarly, Lines 107-108 mention functional data supporting the concepts, but again no references are attributed.
As not only “loss-of-function” mutations can be found in CRC, we have omitted this statement. Furthermore, references have been added as suggested by the reviewer.
Regarding Line 114, there may be new references about the TGF mutation since 1995, which could be included in the text. Additionally, the discussion about ECM and the release of TGFbeta by MMP could be expanded upon, given its relevance to the chosen subject.
Additional references have been added and mechanisms of the release of TGFβ through proteolytic cleavage have been described.
Line 118 raises the question of which tumor suppressors are being referred to, as only the ones listed later in the text are mentioned. It would also be beneficial to readers to elaborate on why the degradation of L-arginine is immunosuppressive in Line 250.
Examples for tumor suppressor genes have been given. Furthermore, the role of L-arginine for T cell function has been described.
Furthermore, Line 226 raises the question of whether "superior predictions" refer to better prognostics.
The statement refers to the fact that the classification of immune infiltration in CRC tissue samples allows a more accurate assessment of the prognosis of individual patients. We agree with the reviewer that the usage of “prediction” in this context is misleading, as it is mostly used in the context of an assessment of a therapeutic response. The sentence has been rephrased to improve clarity and a reference has been added.
Finally, Line 276 would benefit from further explanations and details regarding effector Tregs (eTreg) and their anti-tumor effects in CRC.
The section on the role of Tregs has been expanded and additional references have been cited.
In addition to these points, readers would benefit from a figure summarizing the dual role of TGFbeta, as Figure 1 only depicts its pro-tumoral functions.
The figure and the legend have been changed to include the anti-tumor effects of TGFβ. The title of the figure has been changed to reflect the dual role of TGFβ in the pathogenesis of CRC.
The conclusion section could also be expanded to include a brief summary of the major points that support the dual role of TGFbeta and how pharmacological interference could contribute to cancer treatment.
The conclusion section has been expanded with a more detailed summary of the discussed dual role of TGF signaling in CRC and future directions (see also comments from Reviewer 2).
Finally, I note some minor points that could be addressed in the text.
For instance, in the abstract, line 9 "an important role of" or "the central role of" the tumor microenvironment could be used instead of "a central role for " .
Line 14," please add a comma before "etc".
Line 23 should remove the word "are."
Additionally, in Line 117, the parentheses require clarification, and Line 182 requires punctuation at the end of the sentence.
Thank you for considering these recommendations.
We have changed the text accordingly.
Reviewer 4 Report
This is a potentially interesting review, although I have some issues regarding the reported frequencies of TGBR2 mutations and the description of the CRC oncogenic pathways.
Major issues::
Lines 35-37. “Major pathways involved in the development of CRC include the chromosomal instability pathway (CIN), the microsatellite instability pathway (MSI) and the CpG island methylation pathway [5, 6]”
Despite the review from Schmitt and Gretten that propose these three pathways, the so-called CIMP-high pathway overlaps with MSI for the vast majority of sporadic CRC cases, which are caused by hypermethylation of MLH1, so they are not two independent but actually extremely overlapping pathways. This overlap has been clear from the very first proposal of the methylator phenotype by the group o JP Issa (late 90s), and confirmed by many scientists including the work of Peter Laird unveiling the association of BRAF (early 2000s), hypermethylation of MLH1 and MSI, as well as the later molecular subclassification into CMS in 2015 (reference 10). Later, the work of M. Green’s group revealed that MAFG phosphorylation was involved in the association between BRAF mutation and aberrant hypermethylation of many CIMP-high targets, including MLH1. The molecular basis of the “other CIMP” (CIMP-low) that associates with KRAS mutations has also been explored by the group of Michael Green. The fact of the matter is that for phenotypic and mutational profiling of cancers, the hypermutated (MSI+ and POLE+) vs non-hypermutated cancers provides a better classification than those based on CIMP-, CIMPlow, CIMPhigh. This might be one of the reasons why the TCGA employed that classification in their 2012 CRC marker paper. From the transcriptional point of view, the classification into CMS proposed by Guinney et al seems to be the most consistent, although TGFβ-signalling seems to be a crucial factor for prognosis even within each CMS subgroup.
Lines 39-46. “CIN is the most frequent molecular pathway leading to colorectal cancer. It is characterized by chromosomal changes including aneuploidy, loss of heterozygosity, insertion-deletion mutations and/or amplifications [5]. These mutations usually affect specific oncogenes and/or tumor suppressor genes including the important components of the TGFβ pathway such as TGFβ receptor type 2 (TGFBR2) and SMAD4 (mothers against decapentaplegic homo-logue 4) MSI affects 10-20% of CRCs and arises from mutations in mismatch repair (MMR) genes, which result in mutations of highly repetitive microsatellite regions. Interestingly, TGFBR2 mutations can be found in >80% of CRC cases with high levels of MSI (MSI-H) [7]. Together, TGFBR2 mutations have been identified in ~30% of all CRC cases [8, 9].”
This is very imprecise and should be rewritten.
First, indels do occur in CIN cancers but they are the typical mutation of MSI cancers, at a rate of at least one order of magnitude higher than in MSS (CIN) cancers. Also, TGFBR2 is typically mutated in MSI CRCs, albeit at around 50% and not at the extremely very high frequency >80% stated in this paper, which was based in publications prior to 2003. Please consider more recent and unbiased screenings such as the one published by the TCGA in 2012. Nevertheless, TGFBR2 is rarely mutated in CIN/MSS cancers (<2%, according to the TCGA data).
Moreover, the majority of sporadic MSI cancers are caused by hypermethylation of MLH1, not by mutations in MMR genes. MSI targets all types of microsatellite sequences, not only “highly repetitive regions”. That is precisely the main reason why MSI is oncogenic because, among the many mutations that they accumulate, some take place in unique microsatellite sequences in cancer-related genes such as TGFBR2 or BAX (PMID: 12700659). Frameshift mutations in MSH3/6 occur in these cancers as secondary events because they contain microsatellites in their coding regions making them susceptible to the mutator phenotype already unleashed by epigenetic silencing of MLH1 (PMID: 8700220). Somatic point mutations in MRR genes, in absence of MLH1 hypermethylation, are generally also a secondary event associated with the mutator phenotype driven by POLE mutations.
Lines 319-321. “CAFs develop from various cell types including resident tissue fibroblasts, fibrocytes, bone-marrow derived mesenchymal cells, epithelial cells, endothelial cells, stellate cells and adipocytes [71]”
Although some subpopulations of CAFs originating from different cell types have been described, the majority of CAFs are resident fibroblasts. Also note that those different-origin CAFs populations have been described in other cancer types, not in CRC. I think this should be mentioned to avoid misleading information regarding the origin of CAFs in CRC
In this section, I think it would be appropriate to include the results of PMID: 25706628, reporting that the TGFβ transcriptional signal found in poor-prognosis CRCs originated for the most part from CAFs, not the cancer cells themselves (since most of them are insensitive to TGFβ signaling).
Lines 378-381. “Although previous studies targeting TGFβ in CRC only provided limited results, not at least due to systemic effects, there is growing evidence that the inhibition of the TGFβ pathway in selected patients or in combination with immune checkpoint blockade might be promising strategies.”
The use of “not at least” is very confusing. The limited results are due to systemic effects or not?
Author Response
This is a potentially interesting review, although I have some issues regarding the reported frequencies of TGBR2 mutations and the description of the CRC oncogenic pathways.
Major issues::
Lines 35-37. “Major pathways involved in the development of CRC include the chromosomal instability pathway (CIN), the microsatellite instability pathway (MSI) and the CpG island methylation pathway [5, 6]”
Despite the review from Schmitt and Gretten that propose these three pathways, the so-called CIMP-high pathway overlaps with MSI for the vast majority of sporadic CRC cases, which are caused by hypermethylation of MLH1, so they are not two independent but actually extremely overlapping pathways. This overlap has been clear from the very first proposal of the methylator phenotype by the group o JP Issa (late 90s), and confirmed by many scientists including the work of Peter Laird unveiling the association of BRAF (early 2000s), hypermethylation of MLH1 and MSI, as well as the later molecular subclassification into CMS in 2015 (reference 10). Later, the work of M. Green’s group revealed that MAFG phosphorylation was involved in the association between BRAF mutation and aberrant hypermethylation of many CIMP-high targets, including MLH1. The molecular basis of the “other CIMP” (CIMP-low) that associates with KRAS mutations has also been explored by the group of Michael Green. The fact of the matter is that for phenotypic and mutational profiling of cancers, the hypermutated (MSI+ and POLE+) vs non-hypermutated cancers provides a better classification than those based on CIMP-, CIMPlow, CIMPhigh. This might be one of the reasons why the TCGA employed that classification in their 2012 CRC marker paper. From the transcriptional point of view, the classification into CMS proposed by Guinney et al seems to be the most consistent, although TGFβ-signalling seems to be a crucial factor for prognosis even within each CMS subgroup.
We thank the reviewer for these detailed comments. The section has been completely rewritten based on the TCGA data. It is now mainly focusing on the differentiation of non-hypermutated and hypermutated CRC.
Lines 39-46. “CIN is the most frequent molecular pathway leading to colorectal cancer. It is characterized by chromosomal changes including aneuploidy, loss of heterozygosity, insertion-deletion mutations and/or amplifications [5]. These mutations usually affect specific oncogenes and/or tumor suppressor genes including the important components of the TGFβ pathway such as TGFβ receptor type 2 (TGFBR2) and SMAD4 (mothers against decapentaplegic homo-logue 4) MSI affects 10-20% of CRCs and arises from mutations in mismatch repair (MMR) genes, which result in mutations of highly repetitive microsatellite regions. Interestingly, TGFBR2 mutations can be found in >80% of CRC cases with high levels of MSI (MSI-H) [7]. Together, TGFBR2 mutations have been identified in ~30% of all CRC cases [8, 9].”
This is very imprecise and should be rewritten.
First, indels do occur in CIN cancers but they are the typical mutation of MSI cancers, at a rate of at least one order of magnitude higher than in MSS (CIN) cancers. Also, TGFBR2 is typically mutated in MSI CRCs, albeit at around 50% and not at the extremely very high frequency >80% stated in this paper, which was based in publications prior to 2003. Please consider more recent and unbiased screenings such as the one published by the TCGA in 2012. Nevertheless, TGFBR2 is rarely mutated in CIN/MSS cancers (<2%, according to the TCGA data).
Moreover, the majority of sporadic MSI cancers are caused by hypermethylation of MLH1, not by mutations in MMR genes. MSI targets all types of microsatellite sequences, not only “highly repetitive regions”. That is precisely the main reason why MSI is oncogenic because, among the many mutations that they accumulate, some take place in unique microsatellite sequences in cancer-related genes such as TGFBR2 or BAX (PMID: 12700659). Frameshift mutations in MSH3/6 occur in these cancers as secondary events because they contain microsatellites in their coding regions making them susceptible to the mutator phenotype already unleashed by epigenetic silencing of MLH1 (PMID: 8700220). Somatic point mutations in MRR genes, in absence of MLH1 hypermethylation, are generally also a secondary event associated with the mutator phenotype driven by POLE mutations.
The section has been rewritten according to the reviewer’s comments. The molecular characteristics of hypermutated and non-hypermutated CRC have been described more precisely and more recent numbers on mutations of TGFBR2 have been included.
Lines 319-321. “CAFs develop from various cell types including resident tissue fibroblasts, fibrocytes, bone-marrow derived mesenchymal cells, epithelial cells, endothelial cells, stellate cells and adipocytes [71]”
Although some subpopulations of CAFs originating from different cell types have been described, the majority of CAFs are resident fibroblasts. Also note that those different-origin CAFs populations have been described in other cancer types, not in CRC. I think this should be mentioned to avoid misleading information regarding the origin of CAFs in CRC
In this section, I think it would be appropriate to include the results of PMID: 25706628, reporting that the TGFβ transcriptional signal found in poor-prognosis CRCs originated for the most part from CAFs, not the cancer cells themselves (since most of them are insensitive to TGFβ signaling).
We agree with the comments of the reviewer. The general section on CAFs has been revised to make clear what relates to CRC or not. Furthermore, the already previously integrated study by Calon et al. (PMID 25706628) has been emphasized.
Lines 378-381. “Although previous studies targeting TGFβ in CRC only provided limited results, not at least due to systemic effects, there is growing evidence that the inhibition of the TGFβ pathway in selected patients or in combination with immune checkpoint blockade might be promising strategies.”
The use of “not at least” is very confusing. The limited results are due to systemic effects or not?
The use of “not at least” is definitively confusing. The sentence has been rephrased.